# Combined health risks of cigarette smoking and low levels of physical activity: a prospective cohort study in England with 12-year follow-up

Sarah E Jackson ![ORCID],[1] Jamie Brown,[1] Michael Ussher,[2] Lion Shahab,[1] Andrew Steptoe,[1] Lee Smith[3]

[1]Department of Behavioural Science and Health, University College London, London, UK
[2]Population Health Research Institute, St George's University of London, London, UK
[3]The Cambridge Centre for Sport and Exercise Sciences, Anglia Ruskin University, Cambridge, UK

**Correspondence to**
Dr Sarah E Jackson;
s.e.jackson@ucl.ac.uk

## ABSTRACT

**Objectives** To (1) estimate the combined risks of cigarette smoking and physical inactivity for chronic disease, disability and depressive symptoms and (2) determine whether risks associated with these behaviours are additive or synergistic.

**Design and setting** Longitudinal observational population study using data from Waves 2 (2004/2005) through 8 (2016/2017) of the English Longitudinal Study of Ageing, a prospective study of community-dwelling older adults in England.

**Participants** 6425 men and women aged ≥52 years (mean (SD) 65.88 (9.34) years) at baseline.

**Main outcome measures** Smoking status (never, former, current) and level of physical activity (high, defined as moderate/vigorous physical activity (MVPA) more than once a week; low, defined as MVPA once a week or less) were self-reported at Wave 2 baseline. Self-rated health, limiting long-standing illness, chronic conditions (coronary heart disease (CHD), stroke, cancer, chronic lung disease) and depressive symptoms were reported in each biennial wave.

**Results** Both smoking and low levels of physical activity were associated with increased risk of incident health problems over the 12-year follow-up period. Current smokers with low levels of physical activity had especially high risks of developing fair/poor self-rated health, CHD, stroke, cancer and chronic lung disease compared with highly active never smokers (adjusted relative risk range 1.89–14.00). While additive effects were evident, tests of multiplicative interactions revealed no evidence of large synergistic effects of smoking and low physical activity (Bayes factor range 0.04–0.61), although data were insensitive to detect smaller effects.

**Conclusions** Among older adults in England, there was no evidence of large synergistic effects of smoking and low levels of physical activity on risk of developing chronic disease or depressive symptoms over 12 years. However, additive effects of smoking and low levels of physical activity were evident, underscoring the importance of each of these lifestyle risk behaviours for disease onset.

## Strengths and limitations of this study

► This study represents the first effort comprehensively to examine the combined risks of smoking and low levels of physical activity in a large prospective cohort study.
► Adjustment for a range of relevant covariates took into account potential confounders of the associations between exposures and outcomes.
► Findings were robust to three sensitivity analyses taking different analytical approaches.
► Reliance on self-reported data introduced potential for bias.
► We did not model dynamic effects (ie, the impact of changes in smoking status and physical activity across the time period on disease outcomes) which may have masked some associations.

morbidity and mortality worldwide.[1] Each is associated with substantially increased risk of developing a host of chronic diseases, including coronary heart disease (CHD), cancer and chronic lung diseases.[2–5] Quitting smoking and taking up physical activity leads to improvements in overall health and longevity, even relatively late in life.[2 6–8] Associations between low levels of physical activity and poorer mental health outcomes, including depression and anxiety disorders, have consistently been reported,[9 10] and physical inactivity (defined as not meeting the recommended physical activity guidelines for good health) appears to be causally related to mental health conditions.[3] The evidence on smoking is mixed, with some studies suggesting that the association with poor mental health can largely be explained by common causes, such as genes that predispose to both smoking and depression[11 12] and others finding evidence for a causal relationship.[13 14]

## INTRODUCTION

Smoking and low levels of physical activity are among the leading causes of preventable

The combined health risks associated with smoking and physical inactivity have not been comprehensively examined. This is important because health risk behaviours tend to cluster within individuals.[15–17] Studies in large, representative samples have shown the majority of adults in England and the USA have multiple lifestyle risk factors (eg, smoking, physical inactivity, excessive alcohol intake, low fruit and vegetable consumption),[15 17] and there is evidence to suggest that combinations of lifestyle risk factors have a greater adverse impact on health than would be expected from the added individual effects alone.[18–22] If lifestyle risk factors work synergistically (ie, greater than the sum of the risks associated with each behaviour individually, indicating the behaviours act as effect modifiers for each other) rather than additively (ie, the combined risk is greater than the individual risks associated with each behaviour) to influence disease risk, there may be potential to increase the public health impact of behavioural change interventions by targeting multiple behaviours.[23–25] However, the extant literature on the benefits of multiple behavioural change interventions is mixed, and their effectiveness likely depends on particular behavioural combinations and whether there is genuine synergy between them.[24] A Cochrane review of randomised controlled trials (RCTs) of physical activity in addition to smoking cessation treatment found mixed results, with the majority failing to provide evidence that physical activity aids smoking cessation.[26] However, most of these trials had small samples or a physical activity component insufficiently intense to achieve the desired level of activity.[26] Examination of the risks associated with smoking and physical activity in combination is important to determine synergistic health effects of these risk factors and evaluate the potential usefulness of further research targeting this combination of behaviours in interventions for primary prevention.

There is some evidence from cross-sectional studies to suggest smoking and physical activity interact to influence the risk of adverse physical and mental health outcomes. For example, in a large sample of adult smokers, physical activity was found to moderate the association between nicotine dependence and depression.[27] The results indicated smokers with high nicotine dependence and low physical activity were more likely to be depressed than would be expected on the basis of individual effects of smoking and physical inactivity. Similarly, a survey of undergraduate smokers found that those with a lower level of physical activity had higher odds of depression.[28] However, the cross-sectional study design makes it very difficult to interpret the direction of associations. For example, it is possible being depressed leads to the uptake of smoking and a loss of interest in physical activity, as opposed to being the result of these behaviours. A number of RCTs have examined the impact of physical activity on smoking and cessation outcomes, and provided strong evidence that exercise reduces nicotine cravings and withdrawal symptoms,[26 29 30] although a positive impact on relapse has not clearly been demonstrated.[31] Regarding physical health effects, the evidence is mixed. For example, two small experimental studies have examined the impact of physical activity on cardiovascular biomarkers in smokers and observed improvements in the cardiovascular risk profile over 3 months.[32 33] A cohort study of adults in Copenhagen followed for an average of 11 years found that smokers who engaged in moderate to high levels of regular physical activity experienced a smaller decline in lung function decline and lower chronic obstructive pulmonary disease risk than those who were less active.[34] However, another study that followed middle-aged men in Japan over a similar duration observed no significant interaction between smoking status and level of physical activity for risk of pancreatic cancer.[35] To the best of our knowledge, no studies have evaluated synergistic effects of smoking and physical activity on depression or chronic disease in a large, representative sample using a prospective design.

Using data collected over 12 years from a large population-based sample of older adults living in England, this study therefore aimed to examine the risks of chronic disease and poor mental health associated with cigarette smoking and low levels of physical activity combined. Specifically, we aimed to answer the following research questions:

1. To what extent is the combination of smoking and low physical activity associated with increased risk of the incidence of poor self-rated health, limiting longstanding illness, CHD, stroke, cancer, chronic lung disease and depressive symptoms over 12-year follow-up among older adults who are healthy at baseline, over and above the risks associated with smoking or low physical activity alone, or neither smoking nor low physical activity?
2. Are the combined risks of smoking and low physical activity for these outcomes additive or synergistic?

## METHOD
### Design
This investigation used data from the English Longitudinal Study of Ageing (ELSA) covering a 12-year period. ELSA is a population-based longitudinal panel study of a representative sample of men and women aged 50 and older living in England. The study began in 2002 (Wave 1), with participants recruited from an annual cross-sectional survey of households. Data are collected every 2 years via computer-assisted personal interview and self-completion questionnaires. In alternate (even) waves there is an additional health examination, in which objective measures are obtained. For the present study, baseline data were drawn from Wave 2 (2004/2005; the first wave in which height and weight were measured, allowing inclusion of body mass index (BMI) in the analyses), collected when participants were aged ≥52 years. Follow-up data were collected biennially through to Wave 8 (2016/2017; the most recent wave of available data).

## Measures

### Measurement of exposures

Smoking status was defined as current, former or never smoker on the basis of responses to two yes/no questions: (1) "Have you ever smoked cigarettes?" (2) "Do you smoke cigarettes at all nowadays?" This measure has been validated against salivary cotinine levels in the Health Survey for England.[36]

Physical activity was assessed with three items that asked participants how often they took part in activities that were vigorous (eg, jogging, cycling), moderately energetic (eg, gardening, walking at moderate pace) or mildly energetic (eg, laundry, home repairs). Response options were: more than once a week, once a week, 1–3 times a month, hardly ever/never. Activity examples provided to respondents correspond to metabolic equivalent of task ≥6, ≥3.5 to<6 and ≥2 to<3.5, respectively, for vigorous, moderate and mild activities. For the purpose of analysis, we categorised physical activity into two categories: high physical activity (moderate and/or vigorous activity more than once a week) versus low physical activity (moderate/vigorous activity once a week or less). This measure has been validated against objective, accelerometer-measured hours of moderate–vigorous intensity activity and demonstrates convergent validity in grading a wide range of psychosocial, physical and biochemical outcomes.[7 37–41]

### Measurement of outcomes

We included as outcomes two measures of subjective health (self-rated health, limiting long-standing illness), four diagnosed chronic conditions (CHD, stroke, cancer, chronic lung disease) and one measure of mental health (clinically relevant depressive symptoms).

Self-rated health was assessed using a single item: 'Would you say your health is … poor/fair/good/very good/excellent?' We analysed the proportion of individuals rating their health as fair/poor, as has been done in previous investigations.[42–44]

We also used data on self-reported limiting long-standing illness, which reflects the extent to which participants feel their daily activities are limited by the presence of illness. This was assessed with two questions: (1) 'Do you have any long-standing illness, disability or infirmity? By long-standing I mean anything that has troubled you over a period of time or that is likely to affect you over a period of time.' Those who responded yes were asked: (2) 'Does this illness or disability limit your activities in any way?' Affirmation of a long-standing illness and any form of limitation classified the participant as having a limiting long-standing illness.

Doctor-diagnosed CHD, stroke, cancer and chronic lung disease were self-reported in response to presentation of a list of conditions and the question: 'Has a doctor ever told you that you have (or have had) any of the conditions on this card?'

Depressive symptoms were assessed with an eight-item version of the Center for Epidemiologic Studies Depression Scale, a validated instrument for use in older adults.[45]

Respondents were asked to indicate whether they had experienced depressive symptoms (eg, restless sleep and being unhappy) over the past month using a binary (yes/no) response. Total scores ranged from 0 to 8, with higher scores indicating more depressive symptoms. Data were dichotomised using an established cut-off, with a score of 4 or higher indicating clinically relevant symptomatology.[46]

For each outcome of interest, we analysed the proportion of participants free from that outcome at Wave 2 baseline who reported the presence of the outcome in Wave 3, 4, 5, 6, 7 or 8 (coded 1). Therefore, our dependent variables incorporated all new-onset adverse health outcomes reported by participants across the 12-year follow-up period. For our primary analyses, participants retained in the study at Wave 8 who did not report the presence of the outcome in any wave were coded 0. Participants lost to follow-up before Wave 8 who did not report the presence of the outcome in any wave were coded as missing, because it was not possible to determine their status.

### Measurement of covariates

Demographic variables included baseline age, sex and ethnicity (white vs non-white). Sociodemographic position was indexed using household non-pension wealth, which has been identified as a particularly sensitive indicator in this population.[47] Past-year alcohol intake was categorised as never/rare (never—once or twice a year), regular but infrequent (once every couple of months—twice a week) or frequent (3 days a week—almost every day). BMI was calculated as weight in kilograms/height in metres squared based on objective measurements.

### Statistical analysis

The analysis plan was preregistered on Open Science Framework (https://osf.io/g9p2b/). We amended our prespecified definition of physical activity categories on seeing the distribution of the data, because our original dichotomy of moderate/vigorous physical activity at least once a week resulted in an implausibly high proportion of the sample being classified as high active (~80%). For transparency, results based on the original categorisation are available on Open Science Framework.

We used one-way independent analyses of variance (continuous variables) and Pearson's $\chi^2$ tests (categorical variables) to analyse differences in baseline characteristics by smoking status (never/former/current) and level of physical activity (high/low).

We used log-binomial regression to calculate the relative risks (RRs) with 95% CIs associated with smoking and physical activity of incident fair/poor self-rated health, limiting long-standing illness, CHD, stroke, cancer, chronic lung disease and depressive symptoms over 12-year follow-up among participants who did not report the outcome of interest at baseline. We constructed five models for each outcome. The first and second calculated unadjusted RRs associated with smoking status (reference

category: never smoker) and physical activity (reference category: high active), respectively. The third tested main effects of smoking status and physical activity, and the multiplicative interaction between smoking status and physical activity, controlling for covariates. The fourth and fifth calculated unadjusted and adjusted RRs, respectively, associated with each combination of smoking status and level of physical activity: (1) never smoker/high active (reference category), (2) never smoker/low active, (3) former smoker/high active, (4) former smoker/low active, (5) current smoker/high active and (6) current smoker/low active.

We performed three sensitivity analyses. The first imputed missing outcomes data for those who dropped out of ELSA before Wave 8 and did not report the presence of any of these conditions in their completed waves. A multiple imputation model was run with all exposures and covariates entered as predictors. Five imputed datasets were created, each was analysed separately and the results were combined to produce pooled estimates of effects. The second sensitivity analysis restricted the sample to those with complete data at Wave 2 and Wave 8 to assess healthy survivor effects. The third excluded current smokers who smoke <15 cigarettes per day (indicative of a lower level of nicotine dependence) to address the potential issue of differential rates of smoking cessation in relation to level of physical activity.[48] One would expect a lower rate of successful quitting during the follow-up period among more dependent smokers, so it was thought that excluding those who were less dependent may provide a better reflection of the combined health risks of smoking and low physical activity rather than an artefact of more successful quitting among active smokers generally.

To evaluate the extent to which our data supported the null hypothesis (ie, no synergistic relationship between smoking and physical activity for risk of incident health problems), the experimental hypothesis (ie, synergy between smoking and physical activity) or were insensitive, we calculated Bayes factors (BFs) for the multiplicative interaction results (see online supplementary material for details).

All analyses were conducted in SPSS V.24, with the exception of the BFs which were calculated using an online calculator (http://www.lifesci.sussex.ac.uk/home/Zoltan_Dienes/inference/Bayes.htm).

### Public and patient involvement

No patients were involved in setting the research questions or outcome measures, nor were they involved in the design and implementation of the study. There are no plans to involve patients in dissemination.

### RESULTS
### Sample characteristics

There were 9432 individuals in Wave 2 of ELSA, of whom 7666 (81.3%) participated in the health examination in which objective measurements of height and weight were obtained. We excluded 1241 individuals (16.2%) with missing data, leaving a final sample for analysis of 6425 participants. Compared with those who were excluded, the analysed sample had a similar mean age but were more likely to be male, white and wealthier. They were also more likely to drink alcohol regularly or frequently and had a higher mean BMI, but were less likely to smoke or have low physical activity. The prevalence of chronic disease and depressive symptoms was generally lower in the analysed sample (online supplementary table 1).

Table 1 presents descriptive characteristics measured at Wave 2 baseline overall and by smoking status and level of physical activity. The sample comprised 2902 men and 3523 women aged ≥52 years (mean (SD) 65.88 (9.34) years). Participants were predominantly white (98.8%) and the upper quintiles of wealth were over-represented. The majority (81.1%) reported regular or frequent alcohol intake and the mean BMI was in the overweight range (27.91 (4.87) $kg/m^2$). The prevalence of chronic disease and depressive symptoms ranged from 2.4% (stroke) to 32.9% (limiting long-standing illness).

Some 14.0% of participants were current smokers, 48.9% were former smokers and 37.2% were never smokers. Those who reported current smoking tended to be younger than never/former smokers, and more were from the lower quintiles of wealth. Current and former smokers were more likely than never smokers to be female and white. Former smokers were the most likely to report drinking alcohol frequently and had the highest BMI. Current smokers were the most likely to have low levels of physical activity. They were also more likely than former and never smokers to rate their health as fair or poor, and to report the presence of limiting long-standing illness, diagnosed chronic lung disease and clinically relevant depressive symptoms. Former smokers were the most likely to report CHD and stroke.

Just over a third (34.1%) were classified as having low physical activity. Relative to those with high levels of physical activity, participants with low levels of physical activity were older on average, and a higher proportion were female and from the lower quintiles of wealth (table 1). They were less likely to drink alcohol frequently, had a higher mean BMI and were more likely to be current smokers. Participants with low levels of physical activity were also more likely than those with high levels of physical activity to rate their health as fair or poor, and to report the presence of a limiting long-standing illness, diagnosed CHD, stroke, cancer or chronic lung disease, and clinically relevant depressive symptoms.

### Associations with incident health problems

For each outcome, table 2 summarises the absolute risk and unadjusted and adjusted RRs associated with smoking status and physical activity, and interactions between smoking status and physical activity. Table 3 shows the absolute risk and unadjusted and adjusted RRs associated

**Table 1** Sample characteristics at baseline overall and in relation to smoking status and level of physical activity

| | Whole sample (N=6425) | Smoking status | | | | Physical activity | | |
|---|---|---|---|---|---|---|---|---|
| | | Never smoker (n=2387) | Former smoker (n=3141) | Current smoker (n=897) | P value | High (n=4233) | Low (n=2192) | P value |
| Age in years, mean (SD) | 65.88 (9.34) | 65.55 (9.17) | 66.95 (9.60) | 63.04 (8.12) | <0.001 | 64.77 (8.54) | 68.03 (10.38) | <0.001 |
| Sex, % (n) | | | | | | | | |
| Men | 45.2 (2902) | 34.1 (813) | 53.1 (1667) | 47.0 (422) | <0.001 | 47.5 (2011) | 40.6 (891) | <0.001 |
| Women | 54.8 (3523) | 65.9 (1574) | 46.9 (1474) | 53.0 (475) | – | 52.5 (2222) | 59.4 (1301) | – |
| Ethnicity, % (n) | | | | | | | | |
| White | 98.8 (6345) | 98.1 (2342) | 99.3 (3118) | 98.7 (885) | 0.001 | 98.7 (4180) | 98.8 (2165) | 0.944 |
| Non-white | 1.2 (80) | 1.9 (45) | 0.7 (23) | 1.3 (12) | – | 1.3 (53) | 1.2 (27) | – |
| Wealth quintile, % (n) | | | | | | | | |
| 1 (poorest) | 14.6 (940) | 11.5 (275) | 13.0 (408) | 28.7 (257) | <0.001 | 10.7 (451) | 22.3 (489) | <0.001 |
| 2 | 18.5 (1188) | 17.1 (407) | 17.8 (559) | 24.7 (222) | – | 15.6 (661) | 24.0 (527) | – |
| 3 | 20.8 (1338) | 21.3 (508) | 20.9 (656) | 19.4 (174) | – | 21.4 (905) | 19.8 (433) | – |
| 4 | 22.3 (1432) | 22.9 (546) | 23.7 (743) | 15.9 (143) | – | 24.4 (1034) | 18.2 (398) | – |
| 5 (richest) | 23.8 (1527) | 27.3 (651) | 24.7 (775) | 11.3 (101) | – | 27.9 (1182) | 15.7 (345) | – |
| Alcohol intake, % (n) | | | | | | | | |
| Never/rarely | 18.9 (1213) | 21.5 (513) | 14.9 (468) | 25.9 (232) | <0.001 | 15.4 (651) | 25.6 (562) | <0.001 |
| Regularly | 45.3 (2909) | 48.0 (1145) | 44.3 (1393) | 41.4 (371) | – | 45.0 (1905) | 45.8 (1004) | – |
| Frequently | 35.8 (2303) | 30.5 (729) | 40.8 (1280) | 32.8 (294) | – | 39.6 (1677) | 28.6 (626) | – |
| BMI, mean (SD) | 27.91 (4.87) | 27.84 (4.84) | 28.19 (4.85) | 27.14 (4.93) | <0.001 | 27.48 (4.44) | 28.75 (5.51) | <0.001 |
| Fair/poor self-rated health*, % (n) | 24.5 (1575) | 19.7 (469) | 24.5 (770) | 37.5 (336) | <0.001 | 16.8 (709) | 39.5 (866) | <0.001 |
| Limiting long-standing illness*, % (n) | 32.9 (2111) | 28.0 (668) | 34.5 (1082) | 40.2 (361) | <0.001 | 23.7 (1004) | 50.5 (1107) | <0.001 |
| Coronary heart disease*, % (n) | 8.6 (553) | 6.7 (159) | 10.3 (324) | 7.8 (70) | <0.001 | 6.8 (286) | 12.2 (267) | <0.001 |
| Stroke*, % (n) | 2.4 (152) | 1.6 (39) | 2.9 (91) | 2.5 (22) | 0.009 | 1.5 (64) | 4.0 (88) | <0.001 |
| Cancer*, % (n) | 7.7 (496) | 7.2 (171) | 8.4 (263) | 6.9 (62) | 0.154 | 7.2 (303) | 8.8 (193) | 0.019 |
| Chronic lung disease*, % (n) | 7.3 (466) | 4.4 (105) | 7.7 (241) | 13.4 (120) | <0.001 | 5.4 (228) | 10.9 (238) | <0.001 |
| Clinically relevant depressive symptoms*, % (n) | 13.5 (860) | 11.5 (273) | 13.2 (411) | 19.8 (176) | <0.001 | 9.7 (408) | 20.8 (452) | <0.001 |
| Smoking status, % (n) | | | | | | | | |
| Never smoker | 37.2 (2387) | 100 (2387) | – | – | – | 38.2 (1616) | 35.2 (771) | <0.001 |
| Former smoker | 48.9 (3141) | – | 100 (3141) | – | – | 49.3 (2085) | 48.2 (1056) | – |
| Current smoker | 14.0 (897) | – | – | 100 (897) | – | 12.6 (532) | 16.7 (365) | – |
| Level of physical activity, % (n) | | | | | | | | |
| High | 65.9 (4233) | 67.7 (1616) | 66.4 (2085) | 59.3 (532) | <0.001 | 100 (4233) | – | – |
| Low | 34.1 (2192) | 32.3 (771) | 33.6 (1056) | 40.7 (365) | – | – | 100 (2192) | – |

*Complete data on all health variables at baseline were not a prerequisite for inclusion, so there was a small amount of missing data across these variables. Valid percentages are presented for the ease of interpretation.
BMI, bodymass index.

**Table 2** Main effects of smoking status and physical activity and the interaction between smoking status and physical activity for risks of incident health problems over 12-year follow-up

| | Smoking status | | | Physical activity | | Interaction* | |
|---|---|---|---|---|---|---|---|
| | Never smoker | Former smoker | Current smoker | High active | Low active | Former smoker x low active | Current smoker x low active |
| **Fair/poor self-rated health** | | | | | | | |
| % (n) | 40.9 (529) | 49.0 (744) | 59.2 (225) | 42.3 (999) | 60.0 (499) | – | – |
| RR (95% CI) P value | 1 | 1.20 (1.05 to 1.37) 0.008 | 1.45 (1.19 to 1.76) <0.001 | 1 | 1.42 (1.24 to 1.62) <0.001 | – | – |
| RR$_{adj}$ (95% CI) P value | *1 | 1.14 (0.97 to 1.35) 0.112 | 1.55 (1.22 to 1.99) <0.001 | *1 | 1.19 (0.95 to 1.49) 0.141 | 0.99 (0.74 to 1.34) 0.954 | 1.04 (0.68 to 1.59) 0.846 |
| **Limiting long-standing illness** | | | | | | | |
| % (n) | 57.1 (720) | 62.7 (905) | 64.0 (240) | 57.8 (1345) | 69.2 (520) | – | – |
| RR (95% CI) P value | 1 | 1.10 (0.97 to 1.24) 0.143 | 1.12 (0.93 to 1.35) 0.233 | 1 | 1.20 (1.05–1.37) 0.007 | – | – |
| RR$_{adj}$ (95% CI) P value | 1 | 1.07 (0.93 to 1.24) 0.359 | 1.16 (0.92 to 1.45) 0.205 | 1 | 1.06 (0.86 to 1.31) 0.564 | 0.97 (0.73 to 1.29) 0.845 | 1.08 (0.71 to 1.64) 0.719 |
| **Coronary heart disease** | | | | | | | |
| % (n) | 8.8 (117) | 11.7 (176) | 14.9 (59) | 9.2 (214) | 15.2 (138) | – | – |
| RR (95% CI) P value | 1 | 1.33 (1.04 to 1.70) 0.023 | 1.69 (1.21 to 2.36) 0.002 | 1 | 1.66 (1.32 to 2.08) <0.001 | – | – |
| RR$_{adj}$ (95% CI) P value | 1 | 1.13 (0.82 to 1.55) 0.454 | 1.93 (1.23 to 3.03) 0.004 | 1 | 1.19 (0.78 to 1.82) 0.412 | 1.19 (0.70 to 2.03) 0.521 | 1.15 (0.57 to 2.33) 0.704 |
| **Stroke** | | | | | | | |
| % (n) | 8.2 (113) | 9.9 (159) | 12.9 (54) | 8.1 (196) | 13.3 (130) | – | – |
| RR (95% CI) P value | 1 | 1.22 (0.95 to 1.56) 0.129 | 1.58 (1.12 to 2.22) 0.009 | 1 | 1.64 (1.30 to 2.07) <0.001 | – | – |
| RR$_{adj}$ (95% CI) P value | 1 | 1.13 (0.81 to 1.57) 0.491 | 1.74 (1.06 to 2.85) 0.028 | 1 | 1.40 (0.92 to 2.12) 0.115 | 0.81 (0.47 to 1.40) 0.449 | 1.28 (0.63 to 2.64) 0.496 |
| **Cancer** | | | | | | | |
| % (n) | 13.4 (178) | 15.8 (247) | 17.1 (70) | 13.6 (320) | 18.5 (175) | – | – |
| RR (95% CI) P value | 1 | 1.18 (0.96 to 1.45) 0.112 | 1.28 (0.95 to 1.72) 0.107 | 1 | 1.36 (1.11 to 1.66) 0.002 | – | – |
| RR$_{adj}$ (95% CI) P value | 1 | 1.11 (0.86 to 1.43) 0.445 | 1.44 (0.98 to 2.13) 0.067 | 1 | 1.30 (0.92 to 1.83) 0.139 | 1.01 (0.65 to 1.57) 0.970 | 1.01 (0.54 to 1.88) 0.977 |
| **Chronic lung disease** | | | | | | | |
| % (n) | 3.3 (44) | 7.8 (120) | 20.9 (82) | 5.1 (119) | 13.7 (127) | – | – |

Continued

**Table 2** Continued

| | Smoking status | | | Physical activity | | Interaction* | |
| | Never smoker | Former smoker | Current smoker | High active | Low active | Former smoker × low active | Current smoker × low active |
|---|---|---|---|---|---|---|---|
| RR (95% CI) P value | 1 | 2.34 (1.65 to 3.34) <0.001 | 6.28 (4.28 to 9.21) <0.001 | 1 | 2.67 (2.06 to 3.47) <0.001 | – | – |
| RR$_{adj}$ (95% CI) P value | 1 | 2.77 (1.62 to 4.74) <0.001 | 8.33 (4.62 to 15.00) <0.001 | 1 | 3.50 (1.88 to 6.52) <0.001 | 0.56 (0.27 to 1.16) 0.116 | 0.48 (0.22 to 1.06) 0.070 |
| **Clinically relevant depressive symptoms** | | | | | | | |
| % (n) | 53.5 (418) | 56.4 (535) | 62.4 (181) | 52.8 (714) | 62.9 (420) | | |
| RR (95% CI) P value | 1 | 1.06 (0.90 to 1.24) 0.512 | 1.17 (0.94 to 1.46) 0.168 | 1 | 1.19 (1.02 to 1.39) 0.024 | – | – |
| RR$_{adj}$ (95% CI) P value | 1 | 1.09 (0.90 to 1.34) 0.381 | 1.16 (0.87 to 1.55) 0.309 | 1 | 1.09 (0.84 to 1.41) 0.511 | 0.95 (0.68 to 1.34) 0.782 | 1.07 (0.68 to 1.69) 0.764 |

For each outcome, the sample is restricted to those who did not report the presence of the outcome at baseline. Results therefore indicate the prevalence and RR of new-onset health problems over the follow-up period.

*Multiplicative interaction between smoking status and physical activity. Results can be interpreted as the difference in RR between high active former/current smokers, relative to the difference in RR between high active never smokers. Thus, an interaction term above 1 indicates the disparity between high and low active groups was greater for former/current smokers than never smokers, and an interaction term below 1 indicates the disparity between high and low active groups was smaller for former/current smokers than never smokers. RR, relative risk from bivariate models; RR$_{adj}$, relative risk adjusted for age, sex, ethnicity, wealth, alcohol intake, body mass index and physical activity (for smoking status) or smoking status (for physical activity).

**Table 3** Prevalence and unadjusted and adjusted RRs of incident health problems over 12-year follow-up associated with each smoking/physical activity group

| | Never smoker | | Former smoker | | Current smoker | |
|---|---|---|---|---|---|---|
| | High active | Low active | High active | Low active | High active | Low active |
| **Fair/poor self-rated health** | | | | | | |
| % (n) | 37.0 (354) | 51.9 (48.1) | 44.4 (504) | 62.8 (240) | 52.8 (141) | 74.3 (84) |
| RR (95% CI) P value | 1 | 1.40 (1.13 to 1.75) 0.002 | 1.20 (1.02 to 1.41) 0.027 | 1.70 (1.39 to 2.08) <0.001 | 1.43 (1.13 to 1.81) 0.003 | 2.01 (1.48 to 2.73) <0.001 |
| RR$_{adj}$ (95% CI) P value | 1 | 1.19 (0.95 to 1.49) 0.141 | 1.14 (0.97 to 1.35) 0.112 | 1.35 (1.09 to 1.66) 0.006 | 1.55 (1.22 to 1.99) <0.001 | 1.92 (1.40 to 2.65) <0.001 |
| **Limiting long-standing illness** | | | | | | |
| % (n) | 54.5 (516) | 65.2 (204) | 60.0 (662) | 71.3 (243) | 60.1 (167) | 75.3 (73) |
| RR (95% CI) P value | 1 | 1.20 (0.97 to 1.47) 0.089 | 1.10 (0.95 to 1.27) 0.189 | 1.31 (1.08 to 1.59) 0.007 | 1.10 (0.89 to 1.37) 0.384 | 1.38 (1.00 to 1.91) 0.049 |
| RR$_{adj}$ (95% CI) P value | 1 | 1.06 (0.86 to 1.31) 0.564 | 1.07 (0.93 to 1.24) 0.359 | 1.11 (0.91 to 1.36) 0.321 | 1.16 (0.92 to 1.45) 0.205 | 1.33 (0.96 to 1.85) 0.091 |
| **Coronary heart disease** | | | | | | |
| % (n) | 8.0 (78) | 11.0 (39) | 9.5 (104) | 17.3 (72) | 12.3 (32) | 19.9 (27) |
| RR (95% CI) P value | 1 | 1.38 (0.92 to 2.06) 0.119 | 1.19 (0.88 to 1.62) 0.259 | 2.17 (1.55 to 3.05) <0.001 | 1.53 (0.99 to 2.37) 0.053 | 2.48 (1.55 to 3.99) <0.001 |
| RR$_{adj}$ (95% CI) P value | 1 | 1.19 (0.78 to 1.82) 0.412 | 1.13 (0.82 to 1.55) 0.454 | 1.60 (1.12 to 2.30) 0.011 | 1.93 (1.23 to 3.03) 0.004 | 2.64 (1.59 to 4.37) <0.001 |
| **Stroke** | | | | | | |
| % (n) | 6.7 (67) | 12.0 (46) | 8.9 (103) | 12.6 (56) | 9.7 (26) | 18.5 (28) |
| RR (95% CI) P value | 1 | 1.78 (1.20 to 2.64) 0.004 | 1.33 (0.96 to 1.82) 0.083 | 1.88 (1.30 to 2.73) 0.001 | 1.44 (0.90 to 2.32) 0.128 | 2.76 (1.72 to 4.43) <0.001 |
| RR$_{adj}$ (95% CI) P value | 1 | 1.40 (0.92 to 2.12) 0.115 | 1.13 (0.81 to 1.57) 0.491 | 1.27 (0.86 to 1.89) 0.232 | 1.74 (1.06 to 2.85) 0.028 | 3.12 (1.88 to 5.18) <0.001 |
| **Cancer** | | | | | | |
| % (n) | 12.4 (120) | 16.0 (58) | 14.3 (160) | 19.8 (87) | 15.2 (40) | 20.7 (30) |
| RR (95% CI) P value | 1 | 1.29 (0.92 to 1.81) 0.136 | 1.15 (0.89 to 1.48) 0.276 | 1.60 (1.19 to 2.15) 0.002 | 1.22 (0.83 to 1.79) 0.307 | 1.67 (1.08 to 2.58) 0.022 |
| RR$_{adj}$ (95% CI) P value | 1 | 1.30 (0.92 to 1.83) 0.139 | 1.11 (0.86 to 1.43) 0.445 | 1.45 (1.06 to 1.97) 0.019 | 1.44 (0.98 to 2.13) 0.067 | 1.89 (1.20 to 2.98) 0.006 |
| **Chronic lung disease** | | | | | | |
| % (n) | 1.9 (18) | 7.4 (26) | 5.7 (63) | 13.2 (57) | 15.1 (38) | 31.2 (44) |

Continued

**Table 3** Continued

| | Never smoker | | Former smoker | | Current smoker | |
|---|---|---|---|---|---|---|
| | High active | Low active | High active | Low active | High active | Low active |
| RR (95% CI) P value | 1 | 3.96 (2.14 to 7.31) <0.001 | 3.07 (1.80 to 5.22) <0.001 | 7.07 (4.11 to 12.16) <0.001 | 8.13 (4.56 to 14.49) <0.001 | 16.76 (9.42 to 29.83) <0.001 |
| $RR_{adj}$ (95% CI) P value | 1 | 3.50 (1.88 to 6.52) <0.001 | 2.77 (1.62 to 4.74) <0.001 | 5.42 (3.11 to 9.44) <0.001 | 8.33 (4.62 to 15.00) <0.001 | 14.00 (7.68 to 25.53) <0.001 |
| **Clinically relevant depressive symptoms** | | | | | | |
| % (n) | 50.0 (269) | 61.1 (149) | 53.8 (343) | 61.7 (192) | 57.6 (102) | 69.9 (79) |
| RR (95% CI) P value | 1 | 1.22 (0.95 to 1.57) 0.118 | 1.08 (0.88 to 1.31) 0.470 | 1.24 (0.98 to 1.56) 0.075 | 1.15 (0.87 to 1.53) 0.328 | 1.40 (1.01 to 1.93) 0.042 |
| $RR_{adj}$ (95% CI) P value | 1 | 1.09 (0.84 to 1.41) 0.511 | 1.09 (0.90 to 1.34) 0.381 | 1.14 (0.89 to 1.45) 0.297 | 1.16 (0.87 to 1.55) 0.309 | 1.36 (0.97 to 1.89) 0.071 |

For each outcome, the sample is restricted to those who did not report the presence of the outcome at baseline. Results therefore indicate the prevalence and RR of new-onset health problems over the follow-up period.

RR, relative risk; $RR_{adj}$, relative risk adjusted for age, sex, ethnicity, wealth, alcohol intake and body mass index.

with each combination of smoking status and physical activity.

### Main effects of smoking status

In unadjusted models (table 2), both former and current smokers had significantly higher risks of developing fair/poor self-rated health, CHD and chronic lung disease than never smokers (RR range 1.20–2.34 for former smokers, RR range 1.45–6.28 for current smokers). Risk of stroke was significantly higher among current smokers than never smokers (RR 1.58), but did not differ significantly between former and never smokers (RR 1.22). Smoking status was not significantly associated with the risk of developing a limiting long-standing illness, cancer or clinically relevant depressive symptoms (RR range 1.10–1.28).

After adjustment for age, sex, ethnicity, wealth, alcohol intake, BMI and level of physical activity (table 2), the risk of developing chronic lung disease remained significantly higher among former (adjusted relative risk ($RR_{adj}$ 2.77) and current smokers ($RR_{adj}$ 8.33), and risks of developing fair/poor self-rated health, CHD and stroke were significantly higher among current smokers ($RR_{adj}$ range 1.55–1.93), relative to never smokers. The risk of developing cancer approached statistical significance for current versus never smokers ($RR_{adj}$ 1.44).

### Main effects of physical activity

In unadjusted models (table 2), participants with low physical activity had significantly higher risks of developing fair/poor self-rated health, limiting long-standing illness, CHD, stroke, cancer, chronic lung disease and clinically relevant depressive symptoms than those with high physical activity (RR range 1.19–2.67).

After adjustment for age, sex, ethnicity, wealth, alcohol intake, BMI and smoking status (table 2), the risk of developing chronic lung disease remained significantly higher among those with low versus high physical activity ($RR_{adj}$ 3.50), but other associations were attenuated and became non-significant ($RR_{adj}$ range 1.06–1.40).

### Additive and synergistic effects of smoking status and physical activity

After adjustment for covariates, significant differences in risks of developing fair/poor self-rated health, CHD, stroke, cancer and chronic lung disease were observed across different combinations of smoking status and levels of physical activity (table 3, figure 1).

Relative to never smokers with high physical activity, current smokers with low physical activity had the highest risks of each of these outcomes ($RR_{adj}$ range 1.89–14.00). Risks of fair/poor self-rated health, CHD, stroke and chronic lung disease were also significantly elevated among current smokers with high physical activity ($RR_{adj}$ range 1.55–8.33), and the risk of cancer approached significance ($RR_{adj}$ 1.44), although RRs were lower than those for current smokers with low physical activity.

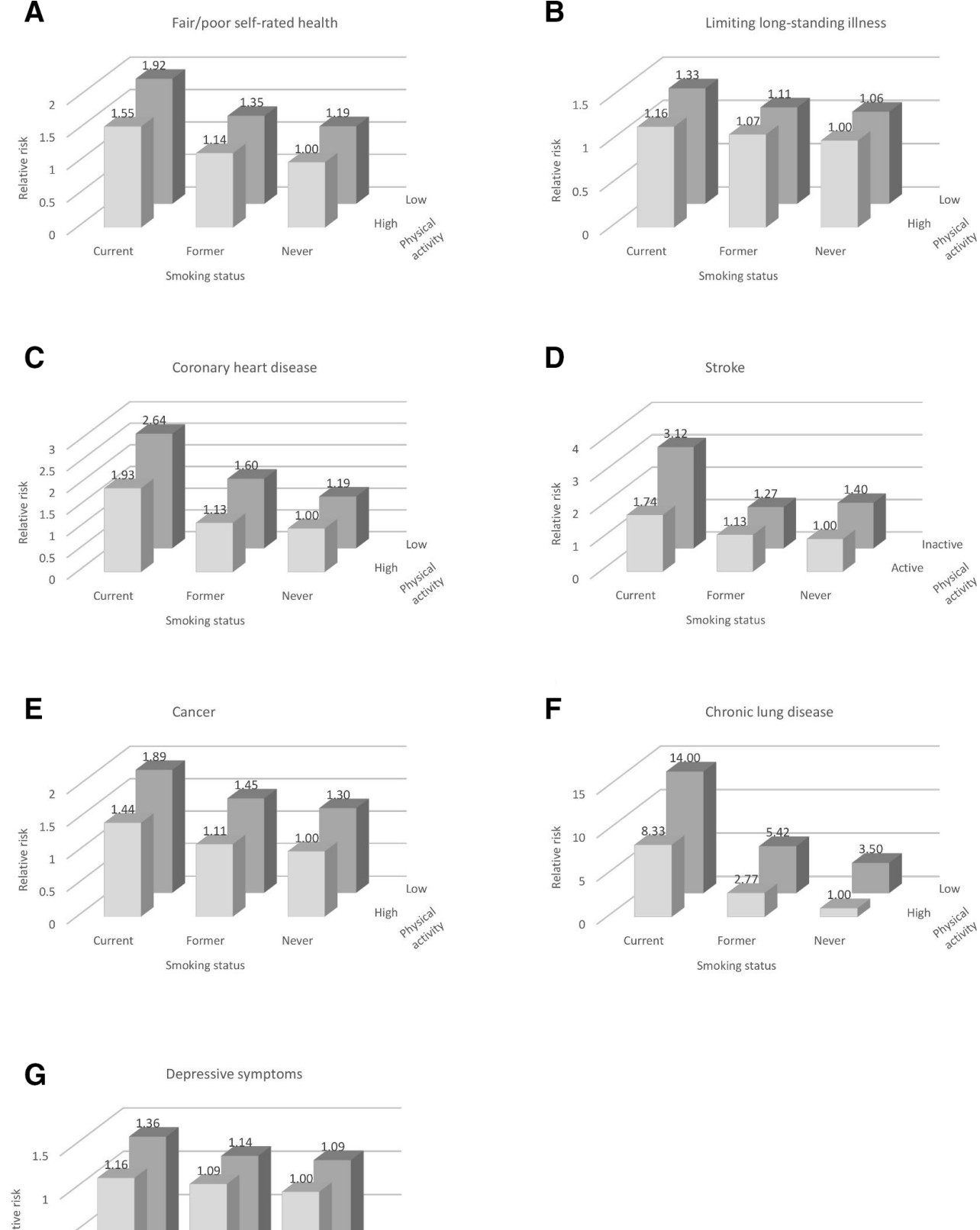

**Figure 1** Relative risks of developing (A) fair/poor self-rated health, (B) limiting long-standing illness, (C) coronary heart disease, (D) stroke, (E) cancer, (F) chronic lung disease and (G) clinically relevant depressive symptoms over 12-year follow-up by baseline smoking/physical activity status, among older adults free of these conditions at baseline.

Risks of fair/poor self-rated health, CHD, cancer and chronic lung disease were also significantly elevated for those with low physical activity who had stopped smoking, although risks relative to never smokers with high physical activity were comparatively lower than were observed for current smokers (RR$_{adj}$ range 1.35–5.42). Chronic lung disease was the only outcome for which significantly elevated risk was observed among former smokers with high physical activity (RR$_{adj}$ 2.77) or never smokers with low physical activity (RR$_{adj}$ 3.50), relative to never smokers with high physical activity.

The risks of limiting long-standing illness and clinically relevant depressive symptoms did not differ significantly across smoking/physical activity groups, although point estimates followed a similar pattern (table 3, figure 1).

While additive effects were evident, with the health risks associated with the combination of current smoking and low physical activity higher than those associated with one or other of these behaviours in isolation (table 3, figure 1), tests of multiplicative interactions revealed no evidence of synergistic effects of smoking and low physical activity (table 2). The only outcome for which the interaction approached statistical significance was chronic lung disease (p=0.070), where the effect was in the opposite direction to what we hypothesised, that is, relative to never smokers, the increase in risk associated with inactivity appears smaller in current smokers.

BFs based on large synergistic effects between smoking status and physical activity indicated the data provided strong evidence for the null hypothesis for chronic lung disease and moderate evidence for the null hypothesis for incident fair/poor self-rated health, limiting long-standing illness, cancer and depressive symptoms, but were insensitive to detect large effects for CHD and stroke (see online supplementary table 2). BFs based on medium and small synergistic effects favoured the null hypothesis but indicated the data were insensitive for all outcomes except chronic lung disease.

### Sensitivity analyses

Sensitivity analyses taking three different analytical approaches produced a very similar pattern of results (see online supplementary material, tables 3–5 and figures 1–3 for full details).

### DISCUSSION

In this large prospective study of older adults, we examined the risks of incident self-rated health, limiting long-standing illness, CHD, stroke, cancer, chronic lung disease and depressive symptoms over 12-year follow-up associated with smoking and low levels of physical activity among individuals free of these conditions at baseline. We observed additive effects of smoking and low physical activity on these outcomes, with older adults who reported both current smoking and low physical activity at higher risk of developing these conditions than those who engaged in one or neither of these lifestyle risk

behaviours. However, there was no evidence of synergistic effects of smoking and low physical activity on the incidence of these conditions.

It has been proposed that targeting multiple behaviours could increase the public health impact of behavioural change interventions,[23–25] but evidence on the effectiveness of this strategy is inconsistent.[24] For example, studies focusing on physical activity and diet have shown interventions that focus on a single behaviour are more effective in increasing the target behaviours, while those that target both behaviours result in greater weight loss.[49] Dieting while trying to stop smoking is associated with worse smoking outcomes,[50] and it is generally recommended smokers do not attempt to diet until several months after quitting.[50] It is likely that the effectiveness of multiple behavioural change interventions relies on there being a synergistic relationship between the target behaviours. The failure of the present study to find evidence of synergy between smoking and low physical activity on risk of chronic disease and depressive symptoms suggests targeting this combination of behaviours is unlikely to be more effective in reducing the risk of these adverse health outcomes than focusing on each behaviour separately. This is consistent with findings of RCTs that have examined effects of physical activity as an adjunct to smoking cessation treatment. A 2014 Cochrane review[26] identified 20 RCTs (total n=5870) that compared an exercise-only intervention or a combined exercise and smoking cessation intervention with a cessation only intervention. Just 2 of the 20 trials found a beneficial effect of including an exercise component on long-term cessation.[26]

However, despite the lack of evidence for synergy between these behaviours, there are other reasons why targeting smoking and physical activity in a multiple behavioural change intervention may be beneficial. For example, changes in physical activity as a result of an intervention may interact differently with smoking compared with more spontaneous changes in physical activity (as reported in cohort studies) and especially so if the intervention is actively used to promote cessation (eg, as a means for reducing cigarette cravings[30]). It is also possible that smoking and physical activity may interact in different ways depending on the timing of changes in the two behaviours.[51]

While the present results provide no evidence for synergistic effects of smoking and low physical activity on health, there were clear additive effects. Current smokers were at higher risk of incident health problems than former or never smokers. People with low physical activity were at higher risk of incident health problems than those who engaged in regular moderate/vigorous intensity physical activity. The combination of current smoking and low physical activity conferred the highest risk of each outcome: notably, individuals who reported both behaviours had more than twice the risk of developing CHD, three times higher risk of having a stroke and 14 times higher risk of developing chronic lung disease over 12-year follow-up than never smokers who

engaged in regular physical activity. These results emphasise the importance of promoting both abstinence from smoking and regular physical activity, and intervening to encourage behavioural change for people with unhealthy lifestyles.

This study had several strengths. The sample was drawn from a large, nationally representative cohort of older adults. The prospective design facilitated assessment of the temporal relationship between smoking and physical inactivity and future disease onset. Assuming the health risk behaviours have a cumulative (dose–response) effect on health outcomes, the older age of the sample meant we had a better chance of detecting an effect given longer exposure in this population group. Adjustment for a range of relevant covariates took into account potential confounders of the associations between exposures and outcomes. Findings were robust to three sensitivity analyses taking different analytical approaches.

There were also a number of limitations. First, the items used to assess smoking status did not specify regular smoking, meaning the group of former smokers encompassed a wide range of smoking histories, from very occasional use to heavy smoking. As such, our results may underestimate the health risks associated with former (regular) smoking. Second, physical activity was self-reported, introducing scope for bias. A recent study documented notable discrepancy between objective measures and self-reports of physical activity, including an age-related decline in activity levels captured by accelerometery that was not observed in self-reports.[52] In addition, levels of physical activity were dichotomised for analysis, distinguishing between those who engaged in moderate or vigorous activities more than once a week and those who engaged in less frequent moderate or vigorous activities. Replication of these analyses using a more objective and detailed measure of physical activity would be useful in validating our results. Third, chronic disease outcomes were based on self-reports of doctor diagnosis, and it is possible some may have been forgotten or not reported. However, validation studies comparing self-reports against medical records generally show high agreement.[53] Fourth, while we included participants who reported the onset of health problems in any wave, regardless of whether they were retained in ELSA through to final follow-up at Wave 8, we excluded from our primary analyses those who did not report health problems or depressive symptoms prior to dropout. This group likely included individuals suffering from the conditions we were studying, but who died before the diseases were identified or could be reported in an ELSA interview. As such, our results may underestimate the impact of our exposures on the health outcomes of interest, although a sensitivity analysis based on imputed data produced similar estimates of associations. There were several differences between the analysed sample and participants we excluded, with the analysed sample generally more advantaged, healthier and less likely to smoke or have low levels of physical activity. As such, our results may not generalise to the entire older population in England. Insofar that a synergistic effect of smoking and low physical activity is greater in less advantaged groups, then the current study could have underestimated the overall effect. In addition, we did not model dynamic effects (ie, the impact of changes in smoking status and physical activity across the time period on disease outcomes) which may have masked some associations, although previous analyses of the ELSA cohort suggest that smoking status and level of physical activity remain stable across waves for the majority of participants.[54] Fifth, although we had a large sample, the number of incident diagnoses was relatively small meaning we likely lacked statistical power to detect significant effects. Indeed, BFs indicated that while the data supported the null hypothesis (ie, no synergistic effects of smoking and physical activity), there was some data insensitivity which meant we were unable to rule out small-sized and medium-sized effects. The small number of incident cases also meant we were unable to conduct more fine-grained analyses, for example focusing on specific cancer types (eg, lung, breast, colorectal) that might be affected by the exposures. Finally, while we adjusted for a range of potential confounders, there were no data available on substance misuse (aside from alcohol intake, which we controlled for) or diet quality. These variables have been associated to varying degrees with our exposures[15–17 55] and outcomes of interest.[56–59] Further research is required to validate our findings with adjustment for these variables. There is also potential for residual confounding by socioeconomic position if there were aspects of this that were not accounted for by our adjustment for non-pension wealth.

## CONCLUSIONS

The present results are not suggestive of large synergistic effects of smoking and low levels of physical activity on risk of developing chronic disease or clinically relevant depressive symptoms (although smaller synergistic effects cannot be ruled out). However, additive effects of smoking and low activity were evident, underscoring the importance of each of these behaviours for disease onset.

**Contributors** SEJ and LeS conceived the study. JB, MU, LiS and AS contributed to the study design. SEJ analysed and interpreted the data, and drafted the manuscript. All authors revised the manuscript critically for intellectual content, and read and approved the final manuscript.

**Funding** This work was supported by Cancer Research UK (C1417/A22962) and the Economic and Social Research Council (ES/R005990/1).

**Competing interests** JB has received unrestricted research funding from Pfizer, who manufacture smoking cessation medications. LiS has received a research grant and honoraria for a talk and travel expenses from manufacturers of smoking cessation medications (Pfizer and Johnson & Johnson).

**Patient consent for publication** Obtained.

**Ethics approval** Ethical approval for the English Longitudinal Study of Ageing was provided by the London Multi-Centre Research Ethics Committee (MREC/01/2/91).

**Provenance and peer review** Not commissioned; externally peer reviewed.

**Data availability statement** Data are available in a public, open access repository.

**ORCID iD**
Sarah E Jackson http://orcid.org/0000-0001-5658-6168

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
