## [Reviewer comments · BMJ Open]

ARTICLE DETAILS

TITLE (PROVISIONAL)	The combined health risks of cigarette smoking and low levels of physical activity: a prospective cohort study in England with 12-year follow-up
AUTHORS	Jackson, Sarah; Brown, Jamie; Ussher, Michael; Shahab, Lion; Steptoe, Andrew; Smith, Lee

VERSION 1 - REVIEW

REVIEWER	Maki Inoue-Choi National Cancer Institute, United States
REVIEW RETURNED	12-Aug-2019

GENERAL COMMENTS	This manuscript described a prospective analysis of cigarette smoking, physical activity, and their interaction on subjective, diagnosed, and mental health outcomes in a large population study among older adults in England. They concluded that cigarette smoking and low physical activity were associated with subsequent chronic diseases and depressive symptoms, and there were additive interactions between smoking and low physical activity; but there was no evidence suggesting synergetic effects between smoking and low physical activity. The manuscript is relatively well-written, and the analyses were well documented. However, I have a few questions about the analysis. I understand that the authors had a sample size issue, but a physical activity variable was very simple, and I think physical activity should be more thoroughly assessed and categorized to make any conclusion on its effect or interaction with smoking on health outcomes. Major comments 1. The authors' discussion on previous studies on smoking and physical activities described in the introduction focus mainly on mental health/depression and not so relevant to chronic diseases. I suggest previous studies on chronic diseases should be expanded, and at least the introduction should be more balanced among outcomes.2. A binary categorization of physical activity level is too simple to capture real physical activity level which may affect health outcomes. Some of the physical activities categorized as "moderately energetic", such as laundry and home repair are questionable. High vs low physical activity with a cut point at more than once or once or less of moderate/vigorous activities – is this based on a physical activity guidelines or other evidence-based criteria? The cut point seems so low, although the authors did find associations with the outcomes.
---

	3. The outcome description is in the supplemental material, but a limiting long-standing illness is not clear and so should be described in the methods. 4. The analysis was adjusted for wealth, but was education level available? If so, why did the authors decided not to adjust the analysis for education? 5. Self-reported general health is usually highly correlated with smoking status and also physical activity level. As Table 1 shows, it is correlated with both smoking and physical activity in this study. I recommend the authors should adjust the analyses for general health. 6. I think smoking and physical activity are very complicatedly correlated and their interactions on health outcomes are also complicated. I am not sure if we should conclude that there is not synergetic effect of smoking and physical activity based on the result of simple interaction analysis as this study performed, especially when physical activity was not thoroughly evaluated or categorized. If it was not possible in the current study, it should be included as a limitation.
--	--

REVIEWER	Richard Patterson MRC Epidemiology Unit, University of Cambridge, UK
REVIEW RETURNED	16-Sep-2019

GENERAL COMMENTS	This is a novel and interesting analyses of the joint effects of smoking and physical activity on a range of health outcomes. There are a few minor issues which the authors should address  1. ELSA data may be representative but once the sample is restricted to those who provided complete data at each wave, your sample may be less so. Although over-represented groups are discussed on page 9 lines 26-29 and a sensitivity analysis which accounted for missing data, I would also like to see a comparison between those excluded and the sample. It might also be worth discussing briefly what the representativeness of the sample might mean for generalizability. As although this is mentioned in the limitations the potential impact was not spelled out. 2. It might be worth adding a sentence or two describing the exposures across the population, which is presented in Table 2. I personally found it interesting that smoking status differed across ethnic groups but physical activity did not. 3. As you rightly acknowledge in the limitations section on page 14 line 5-7 you do not model dynamic changes. However, it seems a shame not to take advantage of multiple measures of exposure. Would it at least be possible to examine the stability of these measures over time? For example, of those who reported high physical active in wave 2, how many were consistently highly active. 4. I would prefer to see confidence intervals quoted in the text in addition to point estimates wherever possible. 5. A small correction but there is a missing zero in Table 3. The lower confidence limit for the RRadj among former smokers who were highly active reads "0.8" but I think it should read "0.80". 6. I presume that number of events was insufficient to focus on specific cancer types which might be affected by the exposures, i.e. lung cancer for smoking and breast or colon cancer for physical activity.
---

	7. I think on page 14 lines 15-20 in your limitations section where you discuss potential additional confounders it is worth mentioning potential residual confounding due to socio-economic position as I am skeptical about the ability of your included variables to full account for this.
--	--

VERSION 1 – AUTHOR RESPONSE

Reviewer(s)' Comments to Author:

Reviewer: 1

Reviewer Name: Maki Inoue-Choi

Institution and Country: National Cancer Institute, United States

This manuscript described a prospective analysis of cigarette smoking, physical activity, and their interaction on subjective, diagnosed, and mental health outcomes in a large population study among older adults in England. They concluded that cigarette smoking and low physical activity were associated with subsequent chronic diseases and depressive symptoms, and there were additive interactions between smoking and low physical activity; but there was no evidence suggesting synergetic effects between smoking and low physical activity. The manuscript is relatively well-written, and the analyses were well documented. However, I have a few questions about the analysis. I understand that the authors had a sample size issue, but a physical activity variable was very simple, and I think physical activity should be more thoroughly assessed and categorized to make any conclusion on its effect or interaction with smoking on health outcomes.

Major comments

1. The authors' discussion on previous studies on smoking and physical activities described in the introduction focus mainly on mental health/depression and not so relevant to chronic diseases. I suggest previous studies on chronic diseases should be expanded, and at least the introduction should be more balanced among outcomes.

Response: We have expanded our discussion of previous studies to achieve a better balance between studies focusing on depression and those on other chronic diseases:

“Regarding physical health effects, the evidence is mixed. For example, two small experimental studies have examined the impact of physical activity on cardiovascular biomarkers in smokers, and observed improvements in the cardiovascular risk profile over three months (32,33). A cohort study of adults in Copenhagen followed for an average of 11 years found that smokers who engaged in moderate to high levels of regular physical activity experienced a smaller decline in lung function decline and lower COPD risk than those who were less active (34). However, another study that followed middle-aged men in Japan over a similar duration observed no significant interaction between smoking status and level of physical activity for risk of pancreatic cancer (35).”

2. A binary categorization of physical activity level is too simple to capture real physical activity level which may affect health outcomes. Some of the physical activities categorized as “moderately

energetic”, such as laundry and home repair are questionable. High vs low physical activity with a cut point at more than once or once or less of moderate/vigorous activities – is this based on a physical activity guidelines or other evidence-based criteria? The cut point seems so low, although the authors did find associations with the outcomes.

Response: The items used to assess physical activity in ELSA were modified from the Whitehall II Health Questionnaire administered in 1991–93. Examples of activities for each question were those most commonly reported in two population-based cohorts in a similar age group (40–65 years) in the UK arm of the European Prospective Investigation into Cancer (EPIC) cohort and the Ely Diabetes Study. These were categorised according to the activity’s metabolic equivalent (MET) score using a compendium of activity energy costs that was designed to facilitate coding of self-reported activity across studies. Activity examples provided to respondents correspond to MET ≥ 6 , MET ≥ 3.5 to < 6 and MET ≥ 2 to < 3.5 respectively for vigorous, moderate and mild activity.

The activities you mention (laundry and home repair) were examples of mildly energetic activities and did not count towards the moderate/vigorous activity in our physical activity exposure variable. The measurement of physical activity used in ELSA does not map neatly onto physical activity guidelines, but a large number of previous studies in ELSA and other cohorts have defined high vs. low physical activity in a similar way, demonstrating robust associations with a range of health outcomes (for example, see the references we cite in support of this measure: “This measure has been validated against objective, accelerometer-measured hours of moderate-vigorous intensity activity and demonstrates convergent validity in grading a wide range of psychosocial, physical, and biochemical outcomes (7,37–41).”)

3. The outcome description is in the supplemental material, but a limiting long-standing illness is not clear and so should be described in the methods.

Response: We have moved the details of the outcome variables from the supplemental material to the method.

4. The analysis was adjusted for wealth, but was education level available? If so, why did the authors decide not to adjust the analysis for education?

Response: Information on education was available, but wealth is a more sensitive indicator of socioeconomic position in the ELSA sample. We now explain this in the method:

“Sociodemographic position was indexed using household non-pension wealth, which has been identified as a particularly sensitive indicator in this population (47).”

5. Self-reported general health is usually highly correlated with smoking status and also physical activity level. As Table 1 shows, it is correlated with both smoking and physical activity in this study. I recommend the authors should adjust the analyses for general health.

Response: While we acknowledge that self-rated health is highly correlated with smoking status and level of physical activity, we do not see it as a potential confounder of associations between these variables and our outcomes of interest (i.e., poor self-rated health does not in itself cause health problems). Rather, it seems more likely that smoking or low physical activity would cause participants to develop health problems (e.g. heart disease, cancer), which in turn would lead these individuals to

rate their general health more poorly. We therefore do not see a strong rationale for adjusting our models for self-rated health.

6. I think smoking and physical activity are very complicatedly correlated and their interactions on health outcomes are also complicated. I am not sure if we should conclude that there is not synergetic effect of smoking and physical activity based on the result of simple interaction analysis as this study performed, especially when physical activity was not thoroughly evaluated or categorized. If it was not possible in the current study, it should be included as a limitation.

Response: We appreciate the need for careful wording of conclusions. We have edited the conclusion of our abstract to better reflect the results as providing no evidence of large synergistic effects, rather than conclusively ruling them out:

“Among older adults in England, there was no evidence of large synergistic effects of smoking and low levels of physical activity on risk of developing chronic disease or depressive symptoms over 12 years.”

Likewise, throughout the manuscript, we talk in terms of the extent of evidence for or against synergistic effects (using Bayes factors to support these claims), and cautiously conclude that “The present results are not suggestive of large synergistic effects of smoking and low levels of physical activity on risk of developing chronic disease or clinically relevant depressive symptoms (although smaller synergistic effects cannot be ruled out).”

We also acknowledge the limitations of our measure of physical activity in the discussion:

“Second, physical activity was self-reported, introducing scope for bias. A recent study documented notable discrepancy between objective measures and self-reports of physical activity, including an age-related decline in activity levels captured by accelerometry that was not observed in self-reports (52). In addition, levels of physical activity were dichotomised for analysis, distinguishing between those who engaged in moderate or vigorous activities more than once a week and those who engaged in less frequent moderate or vigorous activities. Replication of these analyses using a more objective and detailed measure of physical activity would be useful in validating our results.”

Reviewer: 2

Reviewer Name: Richard Patterson

Institution and Country: MRC Epidemiology Unit, University of Cambridge, UK

This is a novel and interesting analyses of the joint effects of smoking and physical activity on a range of health outcomes. There are a few minor issues which the authors should address

1. ELSA data may be representative but once the sample is restricted to those who provided complete data at each wave, your sample may be less so. Although over-represented groups are discussed on page 9 lines 26-29 and a sensitivity analysis which accounted for missing data, I would also like to see a comparison between those excluded and the sample. It might also be worth discussing briefly what the representativeness of the sample might mean for generalizability. As although this is mentioned in the limitations the potential impact was not spelled out.

Response: We have added a supplementary table comparing included and excluded participants on baseline characteristics, summarising the differences at the start of the results section:

“Compared with those who were excluded, the analysed sample had a similar mean age but were more likely to be male, white, and wealthier. They were also more likely to drink alcohol regularly or frequently and had a higher mean BMI, but were less likely to smoke or have low physical activity. The prevalence of chronic disease and depressive symptoms was generally lower in the analysed sample (Supplementary Table 1).”

We also now comment on the generalisability of our findings in the limitations section of our discussion:

“There were several differences between the analysed sample and participants we excluded, with the analysed sample generally more advantaged, healthier, and less likely to smoke or have low levels of physical activity. As such, our results may not generalise to the entire older population in England. Insofar that a synergistic effect of smoking and low physical activity is greater in less advantaged groups, then the current study could have underestimated the overall effect.”

2. It might be worth adding a sentence or two describing the exposures across the population, which is presented in Table 2. I personally found it interesting that smoking status differed across ethnic groups but physical activity did not.

Response: We have added some additional text describing differences in baseline characteristics by smoking status and physical activity:

“Some 14.0% of participants were current smokers, 48.9% were former smokers, and 37.2% were never smokers. Those who reported current smoking tended to be younger than never/former smokers, and more were from the lower quintiles of wealth. Current and former smokers were more likely than never smokers to be female and white. Former smokers were the most likely to report drinking alcohol frequently and had the highest BMI. Current smokers were the most likely to have low levels of physical activity. They were also more likely than former and never smokers to rate their health as fair or poor, and to report the presence of limiting long-standing illness, diagnosed chronic lung disease, and clinically relevant depressive symptoms. Former smokers were the most likely to report CHD and stroke.

“Just over a third (34.1%) were classified as having low physical activity. Relative to those with high levels of physical activity, participants with low levels of physical activity were older on average, and a higher proportion were female and from the lower quintiles of wealth (Table 1). They were less likely to drink alcohol frequently, had a higher mean BMI, and were more likely to be current smokers. Participants with low levels of physical activity were also more likely than those with high levels of physical activity to rate their health as fair or poor, and to report the presence of a limiting long-standing illness, diagnosed CHD, stroke, cancer, or chronic lung disease, and clinically relevant depressive symptoms.”

3. As you rightly acknowledge in the limitations section on page 14 line 5-7 you do not model dynamic changes. However, it seems a shame not to take advantage of multiple measures of exposure. Would it at least be possible to examine the stability of these measures over time? For example, of those who reported high physical active in wave 2, how many were consistently highly active.

Response: The stability of these measures in the ELSA cohort has previously been reported in detail (see report 'Dynamics of Ageing: Evidence from the English Longitudinal Study of Ageing 2002-2012' https://www.ucl.ac.uk/drupal/site_iehc/sites/iehc/files/elsa_-_wave_6_report.pdf, Table 3.1), so we have not done this again in this paper. However, we now make reference to this in the discussion, allowing the interested reader to look up these results:

"In addition, we did not model dynamic effects (i.e. the impact of changes in smoking status and physical activity across the time period on disease outcomes) which may have masked some associations, although previous analyses of the ELSA cohort suggest that smoking status and level of physical activity remain stable across waves for the majority of participants (54)."

4. I would prefer to see confidence intervals quoted in the text in addition to point estimates wherever possible.

Response: Confidence intervals for each point estimate are reported in Tables 2 and 3. We have not included confidence intervals in the text in order to streamline the reporting of the results. In several instances we report ranges of point estimates and adding confidence intervals would mean writing each result out separately, adding substantially to the length of the results text without any obvious benefit.

5. A small correction but there is a missing zero in Table 3. The lower confidence limit for the RRadj among former smokers who were highly active reads "0.8" but I think it should read "0.80".

Response: Thank you for picking this up. We have double checked and corrected it to 0.81.

6. I presume that number of events was insufficient to focus on specific cancer types which might be affected by the exposures, i.e. lung cancer for smoking and breast or colon cancer for physical activity.

Response: That's correct – the total number of incident cancer cases with all cancer types combined was just 495 so it was not possible to conduct any meaningful analyses (given the breakdown by smoking status and level of physical activity) with a greater level of detail. We now acknowledge this as a limitation:

"The small number of incident cases also meant we were unable to conduct more fine-grained analyses, for example focusing on specific cancer types (e.g. lung, breast, colorectal) that might be affected by the exposures."

7. I think on page 14 lines 15-20 in your limitations section where you discuss potential additional confounders it is worth mentioning potential residual confounding due to socio-economic position as I am skeptical about the ability of your included variables to full account for this.

Response: We have added the following:

“There is also potential for residual confounding by socioeconomic position if there were aspects of this that were not accounted for by our adjustment for non-pension wealth.”

VERSION 2 – REVIEW

REVIEWER	Richard Patterson MRC Epidemiology Unit, University of Cambridge
REVIEW RETURNED	22-Oct-2019

GENERAL COMMENTS	I feel that my points have been adequately addressed.
---